# Preparation and Application of Self-Curing Magnesium Phosphate Cement Concrete with High Early Strength in Severe Cold Environments

**DOI:** 10.3390/ma13235587

**Published:** 2020-12-07

**Authors:** Xingwen Jia, Jiayin Luo, Wenxin Zhang, Jueshi Qian, Junmeng Li, Ping Wang, Maohua Tang

**Affiliations:** College of Materials Science & Engineering, Chongqing University, Shapingba N St, Shapingba District, Chongqing 400045, China; 20152910@cqu.edu.cn (J.L.); 201909131116@cqu.edu.cn (W.Z.); qjs@cqu.edu.cn (J.Q.); 201709131150@cqu.edu.cn (J.L.); p.wang@cqu.edu.cn (P.W.); 201809131150@cqu.edu.cn (M.T.)

**Keywords:** magnesium phosphate cement concrete, emergency repair construction, reaction heat, high early strength, self-curing, severe cold environment

## Abstract

The early strength of magnesium phosphate cement (MPC) decreases sharply in severe cold environments ≤−10 °C, with the 2 h compressive strength falling to 3.5 MPa at−20 °C. Therefore, it cannot be used as a repair material for emergency repair construction in such environments. In this study, MPC is adapted for use in such cold environments by replacing part of the dead-burned magnesia (M) in the mixture with a small amount of light-burned magnesia (LBM) and introducing dilute phosphoric acid (PA) solution as the mixing water. The heat released by the highly active acid–base reaction of PA and LBM stimulates an MPC reaction. Moreover, the early strength of the MPC significantly improves with the increase in the Mg^2+^ concentration and the initial reaction temperature of the MPC paste, which enables MPC hardening in severe cold environments. Although the morphology of the reaction products of the MPC is poor and the grain plumpness and size of the struvite crystals are remarkably reduced, the early strength of MPC prepared in the severe cold environment is close to that of MPC prepared under normal temperature. Furthermore, the increases in the early reaction temperature and early strength of magnesium phosphate cement concrete (MPCC) are significantly improved when the PA concentration in the mixing water and the LBM/M ratio are 10% and 4–6% at −10 °C and 20% and 6–8% at −20 °C, respectively. Moreover, self-curing of MPCC can be realized even at −20 °C, at which temperature the 2 h and 24 h compressive strength of MPCC reach 36 MPa and 45 MPa, respectively.

## 1. Introduction

In cases of natural disasters and severe accidents, it is necessary to quickly repair the resulting damages in important infrastructure such as concrete bridges, roads, airport runways, and other structures to ensure smooth implementation of rescue and relief work. However, the high-level technical requirements and constraints of harsh conditions, particularly in severe cold environments of <−10 °C, generally cause problems in implementing emergency repair construction [1,2]. For example, (i) few repair materials are suitable for severe cold environments, (ii) the mixing water is difficult to rapidly obtain, and (iii) the efficiency of the repair construction is significantly reduced. Of these limitations, adapting the emergency repair materials to extreme cold environments remains a particular challenge [3,4].

Emergency repair materials applied to concrete structures should have rapid setting characteristics, and the early strength should be higher than the critical freezing strength in severe cold environments [5,6]. During winter repair construction, early strength agents or antifreeze agents are generally used to improve the early strength of ordinary concrete [7]. In such cases, however, the complex maintenance measures that must be adopted [8] lead to a significant reduction in the efficiency of rush-repair construction. Although organic repair materials such as epoxy resin can solidify at temperatures of −5 °C to −10 °C with the addition of a low-temperature solidifying agent or by adopting auxiliary heating measures, the curing strength of such materials is insufficient for use in rush-repair construction [9].

To address this limitation, several cementitious materials with higher early strength such as sulfoaluminate cement [10], rapid-hardening Portland cement, alkali-activated cement (AAC) [11], and magnesium phosphate cement (MPC) [12] have been employed as repair materials in severe cold environments.

The 24 h compressive strength of sulfoaluminate cement samples prepared at room temperature and cured at 0 °C is only 10.0 MPa [13], and the 4 h compressive strength of rapid-hardening Portland cement cured at 5 °C is 10.5 MPa [14]. However, these two cements are not able to set and harden normally without applying effective maintenance measures when the temperature is below 0 °C [15,16].

Although AAC sets rapidly and has high early strength, its reaction rate is significantly reduced at low temperatures. Brough reported that AAC specimens cured at 5 °C for one day were difficult to demold [17], whereas the 3 day compressive strength of the AAC mortar cured at 7 °C was only 18.4 MPa [18,19]. The compressive strength of AAC prepared with 3 wt.% CaO at room temperature followed by curing at −10 °C for one day was found to be only 14.0 MPa [20]. Therefore, the early strength of AAC does not meet the requirements for repair materials used in severe cold environments.

The MPC showed higher early mechanical properties than those of the aforementioned cementitious materials even at low temperatures. The setting time of the MPC prepared and cured at 0 °C was about 60 min, and the 3 h and 24 h compressive strengths were 14.4 MPa and 22.1 MPa [21], respectively. Compared with the performances of MPC prepared at 22.8 °C, the setting time of that prepared and cured at 0 °C was prolonged by 48 min, and the 3 h compressive strength increased by 4.9 MPa; however, the 24 h compressive strength decreased by 2.5 MPa [21]. Moreover, the 2 h and 24 h compressive strength of MPC mortar prepared at normal temperature and then cured in a low-temperature box at −20 °C after demolding were 10.0 MPa and 21.0 MPa, respectively [22]. The research results of Li [23] also showed that MPC can set and harden even at −20 °C.

Although the previous results demonstrate that MPC has excellent early mechanical properties even in negative-temperature environments, several problems remain. For example, the MPC was prepared and cast at room temperature, as were the raw materials including the mixing water. These conditions differ significantly from the actual conditions in negative-temperature environments. In addition, the specimens were prepared and placed at room temperature until demolding, after which time they were cured in a low-temperature chamber. Thus, the specimens had higher strength before they were placed in the chamber [22,23].

The setting and hardening speed of MPC is so rapid that the compressive strength of the specimens measured before reaching negative temperatures is significantly higher than the freezing critical strength if the MPC specimens are demolded and then placed in a low-temperature chamber [22,24]. Therefore, the conclusion that MPC cured in a low-temperature chamber still has higher early strength is not consistent with the actual conditions.

In fact, field tests revealed that MPC can set and harden in severe cold environments, although its early strength is insufficient for meeting the requirements of rush-repair construction. However, the composition of MPC is very simple, and its strength can still be maintained and developed under negative temperatures [25]. Therefore, MPC still has the potential to be used in the preparation of repair materials with high early strength for emergency repair construction in severe cold environments.

The simulated test environment differs significantly from the actual rush-repair conditions in severe cold environments [22,23], which makes the test results difficult to apply to actual construction. Therefore, to determine the influence of severe cold on the setting time and mechanical properties of MPC and magnesium phosphate cement concrete (MPCC), these materials in the present study were prepared under actual severe cold temperatures.

Considering the cost of repair materials, the main objective of this study is to prepare MPCC rather than MPC for use in practical engineering. To improve the early strength and to realize the self-curing ability of MPCC in severe cold environments, early strength agents and high-activity magnesia are used as activators to enhance the initial reaction rate of the MPC. To improve the early strength of MPC in severe cold environments, the influences of the MPC content, borax content, water–cement (W/C) ratio, and specimen size on the rise in reaction temperature and the early strength of MPCC are evaluated. Finally, self-curing MPCC with high early strength, of which the 2 h and 24 h compressive strengths are more than 30 MPa and 40 MPa, respectively, is used in emergency repair construction at a temperature of about −20 °C. Research on repair materials with high early strength is beneficial for adapting the results to rush-repair of concrete structures in severe cold environments and is of great significance for ensuring the normal operation and rapid recovery of damaged infrastructure such as road pavement, bridges, and airport runways in cold regions.

## 2. Materials and Methods

### 2.1. Materials

The MPC used in a severe cold environment in this study was composed of dead-burned magnesia (M), light-burned magnesia (LBM), ammonium dihydrogen phosphate (NH_4_H_2_PO_4_, ADP), and borax (Na_2_B_4_O_7_·10H_2_O, B). The calcination temperatures and specific surface areas of the M and LBM were 1700 °C and 1100 °C and 230 m^2^/kg and 480 m^2^/kg, respectively. The chemical compositions of these materials are given in Table 1. Both the ADP and B were of industrial grade with ≥99% purity.

The fine aggregate was composed of river sand with a fineness modulus of 2.9, an apparent density of 2620 kg/m^3^, a mud content of 0.5%, and a moisture content of 0.5%. The coarse aggregate was composed of crushed limestone with a maximum particle size of 25 mm, an apparent density of 2670 kg/m^3^, a mud content of 0.3%, a stone powder content of 1.2%, and a moisture content of 0.5%.

The mixing water used was tap water. The early strength agents added to the mixing water were phosphoric acid (H_3_PO_4_, PA), ethylene glycol [(CH_2_OH)_2_], potassium formate (HCOOK), and potassium chloride (KCl), which also were used as antifreeze materials to reduce the freezing point of the mixing water. The concentration of the PA was 85%. The [(CH_2_OH)_2_] is a colorless, odorless, and viscous liquid that is miscible with water. The HCOOK is a white, solid substance that is easily soluble in water, and is non-toxic and non-corrosive with 99.0% purity. The KCl is a white powder that is soluble in water, with 99.5% purity.

### 2.2. Specimen Preparation

The raw materials were precooled in a low-temperature chamber at (−10 ± 2) °C or (−20 ± 2) °C for 24 h so, that the temperature of the raw materials was as close as possible to the test temperature in the severe cold environment. Unless otherwise noted, the mixing proportions of the MPC were M/P = 4, B/M = 0.05, and water/cement (W/C) = 0.14. The mixing of MPC paste and the casting of specimens were conducted outdoors under ambient temperatures of −20 °C to −10 °C. When preparing the MPCC, the raw materials of MPC were first mixed for 30 s, and the mixing water was added and mixed at a high speed for 30 s. Afterward, the coarse and fine aggregates were added into the MPC paste and mixed for 60 s to 90 s. The rotational speed and the output power of the high-speed agitator were more than 1600 r/min and 2 kW, respectively.

The MPC paste and MPCC mixture were poured into plastic molds and placed outdoors for 2 h to ensure that the specimen temperature reached about 0 °C. Then, the specimens were placed in a low-temperature chamber at (−10 ± 2) °C or (−20 ± 2) °C, respectively, until the test age was reached. 

The volume of the low-temperature chamber was 500 L, which is significantly greater than the size of the concrete specimen. However, the increases in the reaction temperatures of the MPC and MPCC measured in the low-temperature box were much higher than those measured in the actual severe cold environment, because the reaction heat of the MPC was not easily released in the closed low-temperature chamber. In addition, the compressive strength of the MPC cured in the low-temperature box was significantly greater than that in the actual severe cold environment, and the 2 h compressive strength was doubled. Notably, if a low-temperature chamber with a smaller volume of less than 200 L is used, the early strength of the MPC would be higher. Therefore, the specimens with 2 h compressive strength were cured in an outdoor severe cold environment, whereas the specimens prepared for 24 h compressive strength were placed outdoors for 2 h and then cured in a low-temperature chamber until the 24 h age was reached.

### 2.3. Methods

The test methods of fluidity, setting time, and soundness of the MPC followed the Chinese national standard [26]. The setting time of the MPC, which was the initial setting time, was measured using a Vicat apparatus (Chongqing, China). The soundness of the MPC was tested using the boiling method at a boiling time of 4 h. The specimen size of the MPC for the compressive strength test was 40 mm × 40 mm × 40 mm, which followed the test method of strength for cement mortar [27].

The slump and setting time of the MPCC mixture followed the performance test method standard of an ordinary concrete mixture [28]. The specimen size of the MPCC for the compressive strength test was 100 mm × 100 mm × 100 mm, according to the Chinese national standard for testing the physical and mechanical properties of concrete [29]. To prevent the influence of normal temperature on the test results, the specimens were held in an insulation box filled with ice bags in a severe cold environment, and were subsequently moved indoors for the mechanical properties test. 

The increase in the MPC reaction temperature was measured using a temperature recorder (Elitech, RC-4, Shanghai, China). The average value of three specimens, measured at their center points, was taken as the final increase in temperature.

The reactivities of the M and LBM were determined using the citric acid method. The liquid state of that citric acid solution was maintained at a negative temperature by reducing the freezing point of the KCl. The samples of M and LBM, each weighing 5.0 g, were mixed and stirred in 100 mL of a 1.0 mol/L citric acid solution at the set temperature. The pH value of the suspension with reaction time was recorded using a pH meter (Five Easy Plus FE28, Mettler Toledo, Switzerland) with a recording time step of 1 s; the coloration time was also observed and recorded.

Samples prepared for scanning electron microscopy (SEM) and X-ray diffraction (XRD) analyses were broken after reaching the test ages, and the hydration of the samples was terminated immediately using absolute ethanol followed by drying at 40 °C. The samples were treated by spray-gold before the SEM test. The hydration product morphology and interfacial transition zone were observed by SEM (TESCAN VEGA 3 LMH, Prague, Czech Republic). The vacuum degree was less than 0.2 Pa, and the scanning voltage was 20 kV. To determine the phase composition and the type of hydration product of the MPC, XRD (PANalytical X’Pert MRD, Almelo, The Netherlands) was used with a Cu target at a working voltage, current, scanning range, and scanning time of 40 kV, 30 mA, 5°–65°, and 10 min, respectively. The porosity was measured using an Hg porosimeter (PM-60GT, Quantachrome, Boynton, FL, USA) at a pressure range of 0–0.34 MPa and a pore size range of 0.003–1080 µm.

## 3. Results and Discussions

### 3.1. Influences of Early Strength Agents on the Early Strength of MPC

The influence of ambient temperature on the early strength of MPC was extremely strong. Compared with that at normal temperature (20 °C), the 2 h compressive strength of the MPC prepared and maintained in the severe cold environment was significantly reduced (Table 2).

As shown in Table 2, the setting time of the MPC prepared by mixing water at 0 °C compared with that at (20 ± 2) °C was prolonged by 40 min and 75 min at ambient temperatures of (−10 ± 2) °C and (−20 ± 2) °C, respectively, and the 2 h compressive strength values were only 29.7% and 10.9% of that at (20 ± 2) °C. In the severe cold environment, the compressive strength of the MPC with no early strength agents decreased significantly even when the mixing water at normal temperature was used. However, compared with the other inorganic cementitious materials, the MPC was still able to set and harden with no curing measures adopted even in the severe cold environment, and its 2 h compressive strength reached the critical freezing strength. Thus, the MPC shows potential as a rush-repair material with high early strength in severe cold environments.

In the severe cold environment, the early strength of the MPC decreased significantly. Therefore, when using the traditional method, the early strength of MPC might be improved by adding an early strength agent. Such agents, which can significantly reduce the freezing point of the mixing water (Figure 1), are employed to keep the mixing water in the liquid state for long periods even in severe cold environments. The rapid combination of Ca^2+^ with H_2_PO_4_^−^ forms Ca(H_2_PO_4_)_2_ in MPC paste, which leads to an abnormal MPC setting time [30]. Therefore, early strength agents containing Ca^2+^ were not considered in the experiment.

In addition, according to the acid–base reaction principle of MPC and the physical properties of PA, a diluted PA solution with a low freezing point can also be used as mixing water in severe cold environments.

As shown in Figure 1, the freezing point of the mixing water decreased significantly with an increase in the dosage of early strength agents. When the concentrations of ethylene glycol, potassium acetate, potassium chloride, or PA in the mixing water were 40%, 40%, 25%, and 20%, respectively, the freezing point of the mixing water decreased to about −20 °C. The effects of early strength agents on the 2 h compressive strength of the MPC are shown in Figure 2.

As shown in Figure 2, the 2 h compressive strength of the MPC was not significantly improved even when early strength agents such as (CH_2_OH)_2_, HCOOK, or KCl were added to the mixing water, which indicates that these agents have no enhancement effects on the early strength of MPC in severe cold environments. Furthermore, the MPC prepared by mixing water with ethylene glycol was unable to condense within 2 h at −20 °C.

Although the compressive strength of the MPC with a soluble substance as an early strength agent still increased after returning to normal temperature, it decreased significantly in the presence of water. The compressive strength of the MPC specimens prepared and cured at −10 °C for seven days, and then placed under normal temperature for seven days with 0 °C mixing water, decreased 50–60% after soaking in water for three days (Figure 3). Therefore, an easily dissolved early strength agent that does not participate in the acid–base reaction of MPC is not suitable for the preparation of repair materials in severe cold environments because a soluble early strength agent will separate out rapidly and dissolve when the MPC makes contact with water (Figure 4).

However, the compressive strength of the MPC prepared by diluted PA solution was higher than that of MPC prepared with mixing water of normal temperature in the severe cold environment (Figure 3). The compressive strength of the MPC prepared with the diluted solution increased significantly when the ambient temperature returned to normal; the MPC showed good water resistance; and the softening coefficient was greater than 0.85 (Figure 3).

The acid–base reaction rate between PA and M was faster than that between ADP and M. Thus, abundant heat was released quickly when diluted PA was used as mixing water (Equation (1)), which caused the initial temperature of the MPC paste to increase rapidly. Moreover, the dissolution of M and ADP also increased with an increase in the MPC paste temperature, and the reaction rate and early strength of the MPC also increased significantly. The acid–base reaction between MgO and PA resulted in soluble Mg(H_2_PO_4_)_2_ and MgHPO_4_ when excess PA was present [1,31] or insoluble Mg_3_(PO4)_2_ in the case of insufficient PA. The mass ratio of PA in the mixing water to MgO was about 3% and 1.5% at −20 °C and −10 °C, respectively. According to Equation (1), the content of M was obviously excessive, and that of the PA was insufficient. Therefore, the reaction product should be insoluble Mg_3_(PO4)_2_ with a solubility of 2.588 × 10^−4^ g/100 mL (20 °C), which will not affect the water resistance of MPC.
3MgO + 2H_3_PO_4_ = Mg_3_(PO4)_2_ + 3H_2_O
△*_f_H* (kJ/mol): 3 × (−601.6) + 2× (−1279.0) = −3955.35 + 3× (−285.8) + −449.95 (Heat Release)(1)

In the severe cold environment, the early strength of the MPC was remarkably improved when diluted PA was used as mixing water, and its long-term strength showed no significant deterioration. Therefore, diluted PA can be used as mixing water, with appropriate solubility values of 10% and 20% at −10 °C and −20 °C, respectively. However, the early strength of the MPC prepared by diluted PA was still lower, particularly at −20 °C, which does not meet the requirements of early strength of repair materials in rush-repair construction. Therefore, it is necessary to take reasonable measures to improve the early strength of MPC.

### 3.2. Strength Improvement of MPC in Severe Cold Environment

In addition to the mix proportion, the early strength and reaction products of MPC depend on the ambient temperature and increases in the reaction temperature [32]. An appropriate reaction temperature increase is conducive to improving the early strength of the MPC in severe cold environments. Increasing the specific surface area (SSA) of M, adding highly active magnesia, and reducing the B/M ratio or W/C ratio are beneficial for improving the reaction temperature increase and early strength of MPC in severe cold environments [33,34].

For meeting the requirements of the operation time and working performance of repair materials, low B/M and W/C ratios of 0.05 and 0.14, respectively, are more suitable for preparing MPC in severe cold environments. Thus, the influences of the SSA of M and the content of LBM with high activity on the setting time and early strength of the MPC are studied in the following sections. It should be noted that the concentrations of PA in the mixing water were 10% and 20% at ambient temperatures of (−10 ± 2) °C and (−20 ± 2) °C, respectively, whereas the temperature of the mixing water was the same as the ambient temperature.

#### 3.2.1. Setting Time

The SSA of M has a significant influence on the setting time and early strength of MPC [35]. Under normal temperature, the suitable SSA of M is about 200–300 m^2^/kg, and the fluidity and setting time of the MPC decrease significantly when the SSA of M is greater than 300 m^2^/kg [36]. The influences of the SSA of M and the LBM/M ratio on the setting time of the MPC prepared in a severe cold environment are shown in Figure 5.

As shown in Figure 5a, the SSA of M increased from 230 m^2^/kg to 450 m^2^/kg, and the setting time of the MPC decreased by 20 min and 15 min at (−10 ± 2) °C and (−20 ± 2) °C, respectively. The average particle size of M increased with an increase in the SSA; that is, a smaller particle size was related to a faster hydration rate, and thus a more rapid condensation rate of MPC.

In addition to the SSA of M, the activity of magnesia also has a significant effect on the condensation rate of the MPC. As shown in Figure 5b, the setting time of the MPC decreased significantly when a small amount of LBM partially replaced the M. At normal temperature, the fluidity and setting time of the MPC decreased significantly as a result of the rapid reaction speed of the LBM. As a result, the MPC could not be poured successfully even with a small amount of LBM. However, the LBM content had no negative effect on the fluidity of the MPC in the severe cold environment when the LBM/M ratio was less than 8%. Therefore, the setting time of the MPC met the requirements of normal operation of rush-repair materials.

#### 3.2.2. Early Strength

The influences of the SSA of M and the LBM/M ratio on the early strength of MPC are shown in Figure 6 and Figure 7, respectively.

As shown in Figure 6, when the SSA of M increased from 230 m^2^/kg to 400 m^2^/kg, the 2 h compressive strength of the MPC increased by 110.5% and 166.7% at (−10 ± 2) °C and (−20 ± 2) °C, respectively. However, the early strength of the MPC decreased owing to the poor fluidity of the MPC paste when the SSA of M reached 450 m^2^/kg. In the severe cold environment, the early strength of the MPC was still lower even when the SSA of M reached 400 m^2^/kg, particularly at −20 ± 2 °C. The poor grindability of M caused the preparation cost of the MPC to increase significantly when the SSA of M was more than 400 m^2^/kg. In addition, the MPC fluidity decreased significantly with an increase in the SSA of M.

The early reaction rate of the MPC is controlled by the dissolution–diffusion mechanism. The solubility and reactivity of M were significantly reduced in the severe cold environment, which is an important reason for the significant reduction in the early strength of the MPC. The dissolution rate of M decreased significantly at −20 °C even if the SSA of M increased significantly. Moreover, the reaction rate of the MPC was still relatively slow, which resulted in a significant decrease in the early strength of the MPC compared with that at room temperature. In addition, the use of M with high SSA can also lead to significant reductions in the fluidity and setting time of MPC, which is not conducive to improving the workability of the MPCC mixture. Therefore, the method of increasing the SSA of M to improve the early strength of MPC lacks practical value.

As shown in Figure 7, the early strength of the MPC was significantly improved by using a small amount of LBM instead of M in the severe cold environment. When the LBM/M ratio was 4–8%, the 2 h and 24 h compressive strength of the MPC exceeded 30 MPa and 40 MPa, respectively. Considering these results, using a small amount of LBM rather than M is more beneficial for improving the early strength of MPC.

The solubility and hydration activity of M are very low. Even LBM with high activity has a solubility of only 8.6 mg/L at 30 °C; the solubility of M is even smaller, at only 2.9 mg/L at 20 °C [36]. In addition, because the Ca^2+^ ions interfere with the solubility test results of M, it is very difficult to accurately test the M solubility. Therefore, citric acid colorimetry is commonly used to evaluate the effects of environmental temperature on the reaction activity of MgO, which can indirectly reflect the solubilities of M and LBM at low temperature (Figure 8).

As shown in Figure 8, the maximum chromogenic time of M with an SSA of 230 m^2^/kg was 69 min at −20 °C. This value was 33 min 30 s even when the SSA reached 450 m^2^/kg, which is still significantly longer than that at normal temperature. Compared with the chromogenic time of M with higher SSA, that of the LBM was only 5 min 41 s even at −20 °C, which indicates that the LBM has high activity even in severe cold temperatures. Therefore, the Mg^2+^ concentration in the initial reaction stage of the MPC paste increases significantly when using a small amount of LBM instead of M in severe cold environments, which is conducive to accelerating the reaction speed of the MPC. In addition, the curve of the rises in the reaction temperature of the MPC (Figure 9) indicates remarkable increases with an increase in the LBM/M ratio, so that the MPC temperature can be maintained in a positive state within about 3 h. The amount of heat released by the rapid reaction of LBM plays a self-curing role, thus the early strength of the MPC significantly increases even in severe cold environments.

The reaction process of the MPC can be divided into six stages: (i) the ADP hydrolysis period, (ii) MgO dissolution period, (iii) Mg (H_2_O)_6_^2+^ growth period, (iv) struvite growth acceleration period, (v) struvite growth deceleration period, and (vi) struvite growth stable period [30,36,37]. Among them, the first two stages are the initial reaction period of the MPC, and the third stage is the induction reaction period. The reaction rate and temperature increase in the aforementioned three stages determine the formation and growth of struvite, as well as the early strength of the MPC. The growth of struvite crystals depends on the temperature, as do the hydrolysis of ADP and the dissolution rate of MgO. A stronger influence on the reaction temperature increase during the induction period of the MPC results in greater MPC strength development. The ambient temperature has a significant effect on the reaction temperature increase in the induction period of the MPC. The initial temperature of the MPC paste decreased significantly with a decrease in ambient temperature. This resulted in remarkably longer initial reaction and hydration induction periods, which led to a significantly decrease in the early strength of the MPC. Thus, the early strength of the MPC is improved by increasing the hydrolysis rate of ADP and the dissolution of MgO in the initial reaction stage of the MPC.

The pH value of the mixing water decreased significantly when PA was used as an early strength agent. More H^+^ was released from the hydrolysis of PA, which accelerated the dissolution of MgO and resulted in the rapid release of additional Mg^2+^ (Equation (2)). Moreover, abundant heat was released in the acid–base reaction between LBM and PA, which caused the temperature of the MPC paste to increase rapidly. The solubility of ADP and M increased rapidly with an increase in the MPC paste temperature, which accelerated the early reaction process of the MPC (Equation (3)). Furthermore, more heat was released through the MPC reaction, which sustained the reaction. In the normal temperature environment, the large amount of heat led to the formation of a high-temperature MPC matrix, struvite decomposition, and matrix cracking, which reduced the early strength of the MPC. However, in the severe cold environment, the reaction heat stimulated the MPC reaction and played a self-curing role to ensure that a positive temperature state was maintained in the MPC within about 3 h. Moreover, the 2 h compressive strength of the MPC in the severe cold environment was close to that under normal temperature.
MgO + 2H^+^ = Mg^2+^ + H_2_O
△_f_*H*_m_^Ɵ^ (kJ·mol^-1^): −601.70 + 0 = −466.85 + −285.83 + −150.98 (Heat Release)(2)
MgO + NH_4_H_2_PO_4_ + 5H_2_O = MgNH_4_PO_4_·6H_2_O
△*_f_H*^o^(kJ/mol): −601.6 + −1445.1 + -285.8 × 5 = −3681.9 + −206.2 (Heat Release)(3)

#### 3.2.3. Reaction Products

The reaction rate of the MPC was reduced significantly in the severe cold environment, which also adversely affected the formation and morphology of the reaction products. Therefore, the types and morphologies of the MPC reaction products were analyzed with a small amount of LBM partially replacing the M and a diluted PA solution used as mixing water.

An adequate reaction of PA and LBM increased the reaction temperature of the MPC paste, which in turn significantly increased the dissolution and reaction speed of the M and ADP. Therefore, the acid–base reaction speed of the MPC was enhanced, and struvite was produced gradually (Figure 10). In addition, a large amount of heat was released with the rapid reaction of MPC, which enabled the material to maintain a higher early strength even in the severe cold environment.

As shown in Figure 10a, the reaction of MPC at the 24 h age was sufficient. Moreover, the struvite crystals exhibited shapes of lumps or particles with full crystals at a normal temperature of 20 °C. The element proportion of the MPC reaction products at normal temperature was Mg/P/N = 1.09:1.07:1, which indicates struvite products. In the severe cold environment at −20 °C, numerous struvite crystals with poor morphologies and smaller particle sizes were generated with the MPC reaction at the 24 h age, and obvious cracking appeared in the MPC matrix. At this temperature, the element proportion of the MPC reaction products was Mg/P/N = 1.48:1.66:1 (Figure 10b), and the proportion of elemental P was significantly higher than that at normal temperature. These results indicate struvite and residual phosphate as reaction products. The ratio of elemental P was relatively high at −20 °C, which indicates that the early reaction of MPC was insufficient at −20 °C and that some ADP not involved in the reaction of MPC remained (Figure 11).

As shown in Figure 11, the MPC reaction products prepared in the severe cold environment include struvite, which was also identified as a normal-temperature product. However, MgNH_4_PO_4_·H_2_O was not found, nor was Mg_3_(PO_4_)_2_·*x*H_2_O, the product of PA and LBM, because of its amorphous form and low occurrence.

#### 3.2.4. Soundness

The early strength of MPC increased remarkably when the LBM partly replaced the M in the severe cold environment. However, the volume stability of repair materials must also be considered because an insufficient amount of LBM results in cracking and MPC intensity attenuation. Therefore, the effect of the LBM content on the soundness of MPC was analyzed by applying the boiling method (Figure 12).

As shown in Figure 12, no visible deformation, cracks or significant dissolution were noted in MPC samples boiled for 4 h, which indicates that the partial substitution of LBM for M had no adverse effects on the soundness of the MPC. Owing to the high reactivity of LBM, a small amount of this material fully reacted with PA and ADP at the initial MPC reaction stage, which eliminated the MPC soundness problem and any adverse effects on the long-term performance of MPCC. Therefore, a small amount of LBM instead of M was used as an activator to quickly release hydration heat in the initial MPC reaction stage to realize self-curing in the MPCC with high early strength.

### 3.3. Mechanical Properties of MPCC

#### 3.3.1. Early Strength of MPCC

The increased reaction temperature is a key factor in controlling the early strength of MPCC in severe cold environments. Moreover, the MPC dosage, B content, and W/C ratio are main factors in determining the increase in MPCC reaction temperature. Therefore, the influences of the mass ratio of MPC to aggregate (MPC/A) on the early strength of the MPCC are studied in this section, and the test results are shown in Table 3.

As shown in Table 3, the early compressive strength of the MPCC prepared at room temperature was highest at an MPC/A ratio of 1:2. In addition, the early compressive strength of the MPCC prepared at (−10 ± 2) °C and (−20 ± 2) °C decreased with a decrease in the MPC/A ratio. Because the reaction heat of the MPC is absorbed by aggregates, the temperature increased and the early compressive strength of the MPCC decreased with an increase in aggregate content at the same ambient temperature. However, the early strength of the MPCC with same mix proportion was only slightly reduced with a decrease in the ambient temperature. This result indicates that an increase in the early temperature can significantly increase the early strength, even in severe cold environments.

The early strength of the MPCC decreased gradually with an increase in aggregate content at a low temperature. However, the 2 h compressive strength was still greater than that at 30 MPa when the MPC/A ratio was 1:2, which fully meets the requirements of emergency repair construction in severe cold environments. Considering the preparation cost, mixture properties, and early strength of the MPCC, an MPC/A ratio of 1:2 is suitable.

The W/C ratio is another important factor affecting the early strength of the MPCC [38]. The influences of this ratio on the early strength of the MPCC are shown in Figure 13.

As shown in Figure 13, both the 2 h and 24 h compressive strength of the MPCC reached the highest values at a W/C ratio of 0.16 in the severe cold environment. It is generally considered that reducing the W/C ratio is conducive to improving the compressive strength of concrete; however, the influence of mixture fluidity on the compressive strength of concrete should also be considered. At a W/C ratio of 0.14, the fluidity of the MPCC mixture was poor; thus, its compressive strength was lower than that of the MPCC at a W/C ratio of 0.16. The fluidity of the MPCC increased significantly with an increase in the W/C ratio, and the slump flow of the MPCC mixture exceeded 700 mm when the W/C ratio was 0.18, although its compressive strength decreased significantly. At a W/C ratio of 0.22, the MPCC mixture froze, and its compressive strength was significantly reduced because the excessive free water in the mixture did not participate in the MPC reaction. Therefore, the W/C ratio of the MPCC should not be greater than 0.18 in severe cold environments.

The MPC reaction rate was reduced significantly with a decrease in ambient temperature. Therefore, a small amount of B was sufficient for producing a longer MPCC setting time and higher early strength in the severe cold environment (Figure 14).

As shown in Figure 14, the early strength of the MPCC decreased significantly with an increase in B. In particular, the 2 h compressive strength of the MPCC prepared at (−20 ± 2) °C was 13.5 MPa when the B/(M + LBM) ratio reached 7%; when the ratio reached 9%, the specimen froze and the 2 h compressive strength of the MPCC decreased sharply. Therefore, a 3–5% ratio of B/(M + LBM) enables the MPCC to obtain a suitable setting time and high early strength in severe cold environments.

#### 3.3.2. Size Effect

At normal temperature, the size effect of the compressive strength of ordinary concrete means that the test value of the compressive strength gradually decreases with an increase in specimen size [39,40]. However, the reaction speed and temperature increase were the main factors controlling the early strength of the MPCC, particularly in the severe cold environment. The increase in MPCC reaction temperature depended on the specimen size, which also had a significant effect on the early strength of the MPCC in the severe cold environment (Figure 15).

As shown in Figure 15, the early strength of the MPCC increased gradually with an increase in specimen size. The influence of specimen size on the compressive strength was significant, particularly at −20 °C, which indicates that the early strength of the MPCC depends on the increase in reaction temperature. In the severe cold environment, a larger specimen size was related to a greater increase in the reaction temperature (Figure 16) as well as an increase in the early strength of the MPCC. Therefore, self-curing of the MPCC can be realized by the reaction heat released in the early MPC reaction stages when the thickness of the repair location is greater than 70 mm. This enables the 2 h compressive strength of the MPCC without auxiliary curing measures to be greater than 30 MPa at (−20 ± 2) °C.

#### 3.3.3. Long-Term Strength

The growth rate of the compressive strength decreased significantly when the MPCC was cured at room temperature for more than 7 days. Similarly, the compressive strength development ended when the MPCC specimen continued to freeze for more than 3 days in the severe cold environment. Under such conditions, the central temperature of the specimen was usually lower than 0 °C after 3 h; afterward, the reaction rate of the MPC decreased significantly, which led to a gradual decrease in the growth of the MPCC compressive strength. 

However, the compressive strength of the MPCC specimens placed in the severe cold environment increased remarkably after returning to normal temperature. The compressive strength of the MPCC specimens placed in a low-temperature box at (−20 ± 2) °C for 7 days was 53.5 MPa. Afterward, the specimens continued to cure at (20 ± 2) °C until the 28 days age was reached, and the compressive strength of the MPCC increased to 63.0 MPa, which is 77.3% of that of the MPCC cured at room temperature for 28 days. Therefore, the long-term strength of the MPCC will not decrease even if placed in a severe cold environment for long periods.

### 3.4. Engineering Application of MPCC

To further verify the comprehensive performance of the MPCC in an actual repair project in a severe cold environment, repair construction of pavement damage of the Zhuozi Mountain section of the Beijing–Yinchuan highway (G110) in Hohhot City, the capital of the Inner Mongolia Autonomous Region, China, was conducted on 5 January 2020. The repair construction began at 07:00 CST and lasted until 13:00 CST. The area of damaged pavement was about 10 m^2^, and the casting volume of the MPCC was about 1.2 m^3^. To improve the construction efficiency, the raw materials were weighed in advance according to the mix proportion of the MPCC. The procedure of rush-repair construction is shown in Figure 17. The environmental temperatures recorded at the construction site are shown in Table 4, and the mixture performance and early strength of the MPCC are shown in Table 5.

In particular, the following sequence for preparing the MPCC must be adopted in the stirring process of the MPCC mixture in a severe cold environment. First, MPC powder and mixing water are stirred to form a paste; afterward, coarse and fine aggregates are added to the paste. If the MPC powder and aggregates are mixed thoroughly and then combined with the mixing water, some of the MPCC raw materials are unable to be incorporated into the mixture because the low temperature of the aggregates causes part of the mixing water to freeze quickly.

As shown in Table 5, the slump of the MPCC mixture was 210 mm (Figure 18). Thus, it was able to be filled and compacted with no vibration after pouring, which is helpful for simplifying the construction technology and improving the efficiency of emergency repair construction. The repair area of the concrete pavement was significantly larger than the surface area of the 100 mm cube specimen (Figure 18), and the maximum temperature increase of the MPCC measured at the construction site was 36.5 °C. This value is about 3.9 °C lower than the temperature of the cube specimen. The 2 h compressive strength measured using a rebound meter was 30.0 MPa, which is about 5.0 MPa lower than that of the MPCC specimens. Nevertheless, the real difference can be smaller because of errors induced by the rebound method.

In severe cold environments, using a diluted PA solution as the mixing water and partial replacement of M with a small amount of LBM can significantly improve the early reaction temperature increase of the MPCC mixture, accelerate its setting and hardening, and enable its self-curing, so that the 2 h compressive strength of the MPCC prepared at −20 °C was more than 30 MPa. The 2 h compressive strength reached 30 MPa, which enables rapid recovery of roads or airport runways 2 h after the completion of the emergency repair construction. After nine months of service, the repaired pavement essentially remained in good condition (Figure 19). This result demonstrates that self-curing MPCC with high early strength can be used for emergency repairs of roads and airport runways in severe cold environments.

## 4. Conclusions

The early strength of the MPC is insufficient for meeting the requirements of emergency repair engineering in severe cold environments. The reaction temperature increases of MPC paste increased significantly through the rapid reaction of diluted PA with a small amount of LBM. This stimulated the initial reaction of the MPC and helped to achieve self-curing of the MPCC with high early strength to be used for emergency repair materials in severe cold environments. The main results of this study and the conclusions are summarized in the following points.
The condensation rate and early strength of the MPC were significantly reduced in the severe cold environment. Commonly used early strength agents that are not involved in the acid–base reaction of the MPC cannot significantly improve the early strength of the MPC and lead to remarkable decreases in the long-term strength and water resistance of the MPC.Compared with the addition of diluted PA or a small amount of LBM alone, a higher early strength of the MPC prepared by simultaneous addition of diluted PA and LBM instead of M was obtained even in the severe cold environment. This result indicates that the reaction heat released in the initial reaction period determines the early strength of the MPC.The heat released by the rapid reaction of LBM and diluted PA played a role in stimulating the initial reaction of the MPC in the severe cold environment. The 2 h and 24 h compressive strengths of MPC with LBM/M ratios of 4–6% at −10 °C and 6–8% at −20 °C reached 30 MPa and 40 MPa, respectively.The MPC reaction was still insufficient even with the help of early reaction heat released by the diluted PA and LBM in the severe cold environment. Moreover, the plumpness and sizes of struvite particles with a poor morphology were reduced compared with those at normal temperature, although the early strength of the MPC was close to that at normal temperature.The increase in the early reaction temperature of the MPCC mixture was significantly improved when diluted PA solution was used as the mixing water and a small amount of LBM partially replaced the M for preparing the MPCC in the severe cold environment. These measures enabled self-curing of the MPCC. The suitable mix proportions of the MPCC were found to be MPC/A = 1:1–1:2, B/M = 0.03–0.05, and W/C = 0.14–0.16, and the 2 h and 24 h compressive strength values of the MPCC reached 36.0 MPa and 45.0 MPa, respectively, at −20 °C.

## Figures and Tables

**Figure 1 materials-13-05587-f001:**
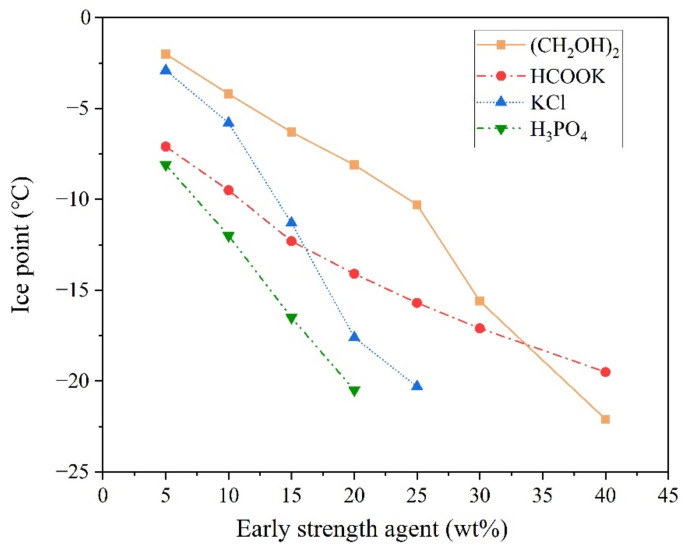
The freezing point of mixing water containing antifreeze agents.

**Figure 2 materials-13-05587-f002:**
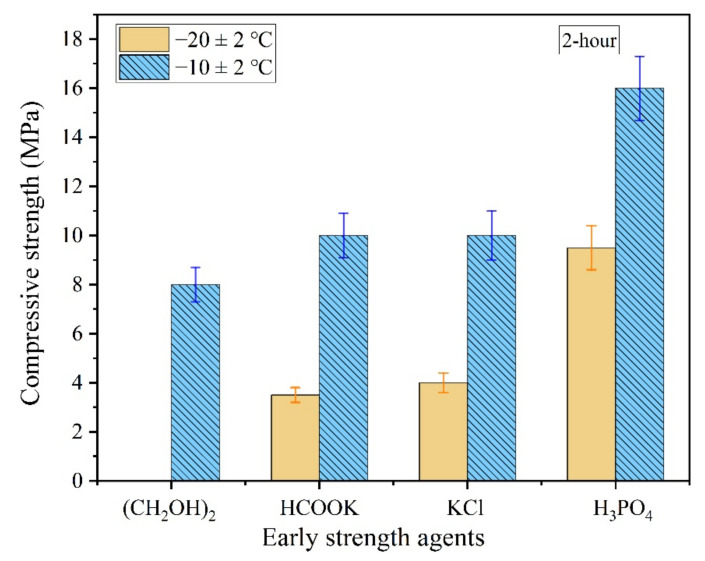
The 2 h compressive strength of the magnesium phosphate cement (MPC) with early strength agents in severe cold environment.

**Figure 3 materials-13-05587-f003:**
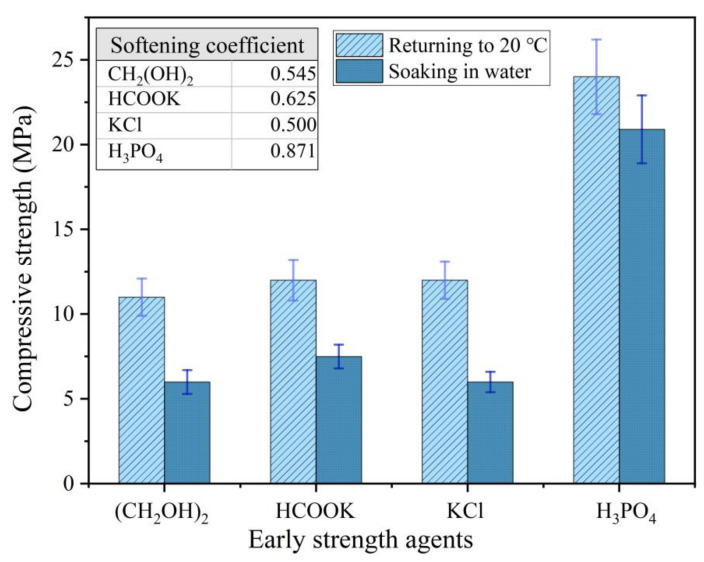
The strength deterioration of the MPC with early strength agents after soaking in water.

**Figure 4 materials-13-05587-f004:**
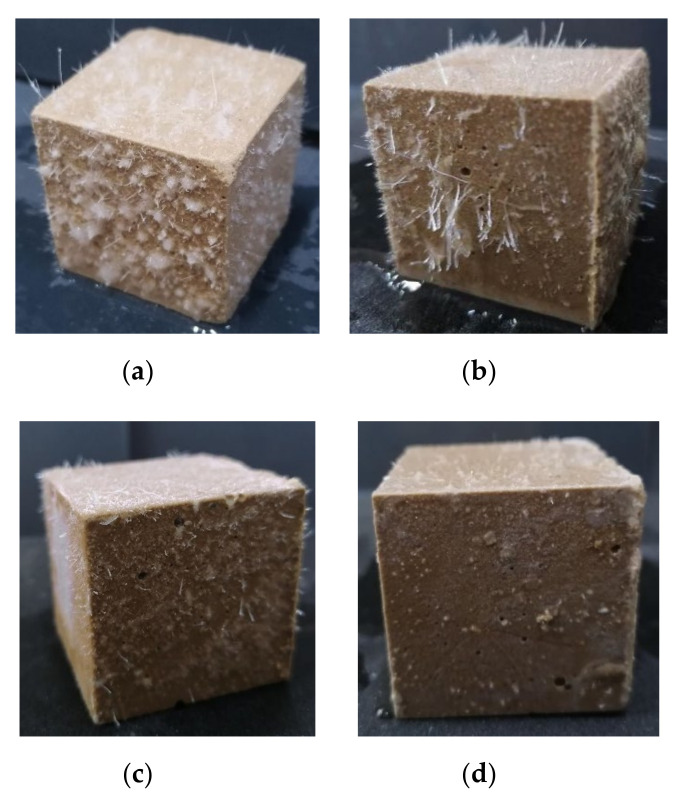
The surface state of MPC specimens prepared with early strength agents after soaking in water: (**a**) (CH_2_OH)_2_; (**b**) HCOOK; (**c**) KCl; and (**d**) H_3_PO_4_.

**Figure 5 materials-13-05587-f005:**
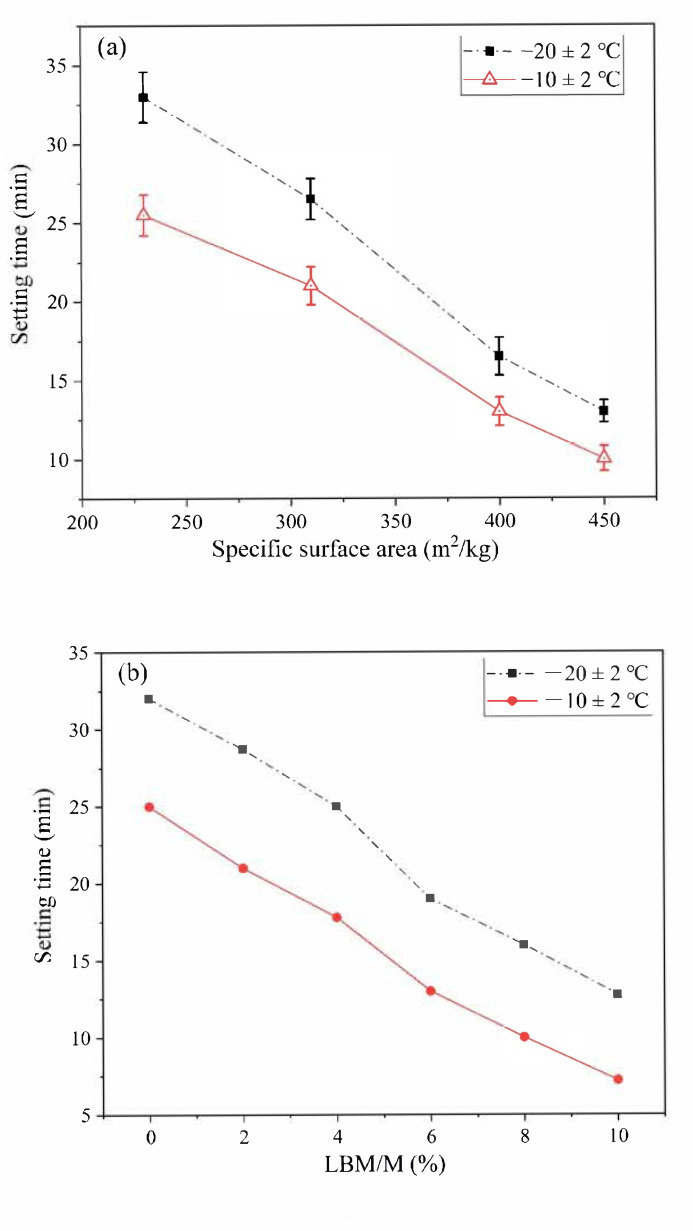
The influences of specific surface area (SSA) of magnesia (M) and light-burned magnesia (LBM)/M ratios on the setting time of the MPC: (**a**) SSA of M; (**b**) LBM/M ratio.

**Figure 6 materials-13-05587-f006:**
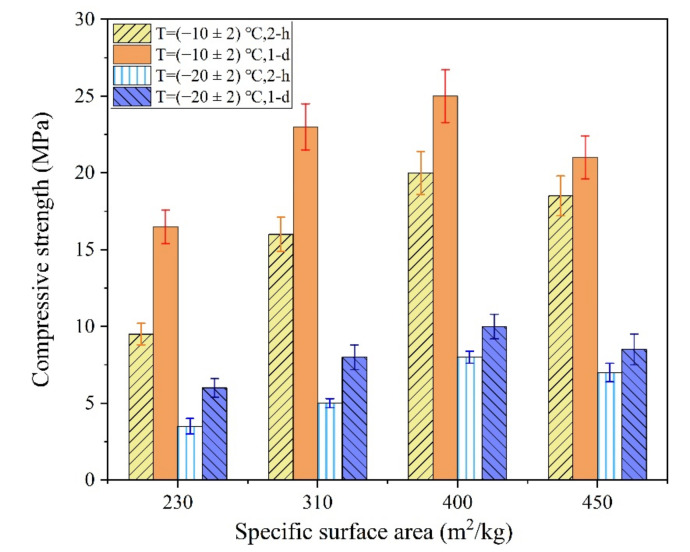
The influences of SSA of M on the early strength of the MPC.

**Figure 7 materials-13-05587-f007:**
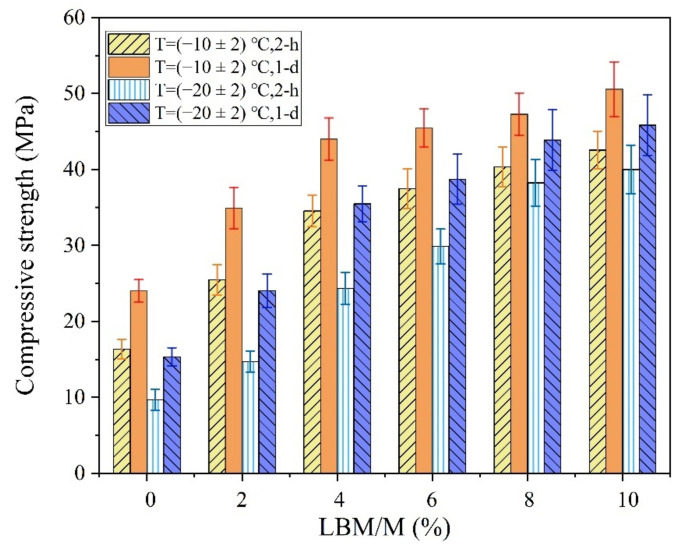
The influences of LBM/M ratios on the early strength of the MPC.

**Figure 8 materials-13-05587-f008:**
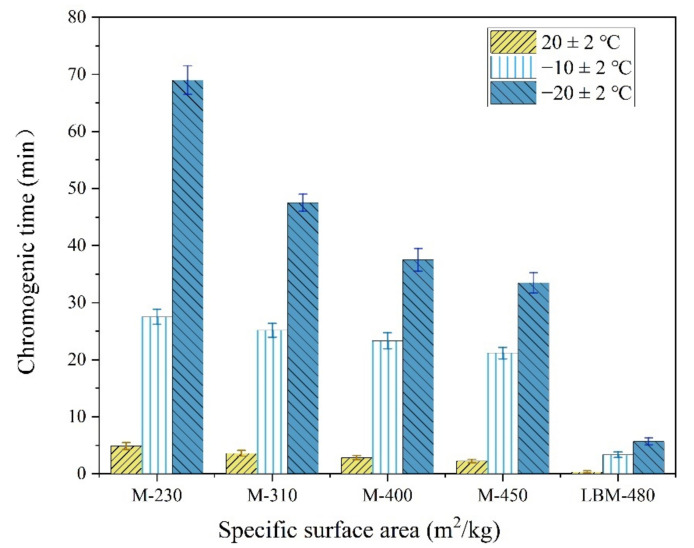
The chromogenic time of MgO determined by the citric acid method.

**Figure 9 materials-13-05587-f009:**
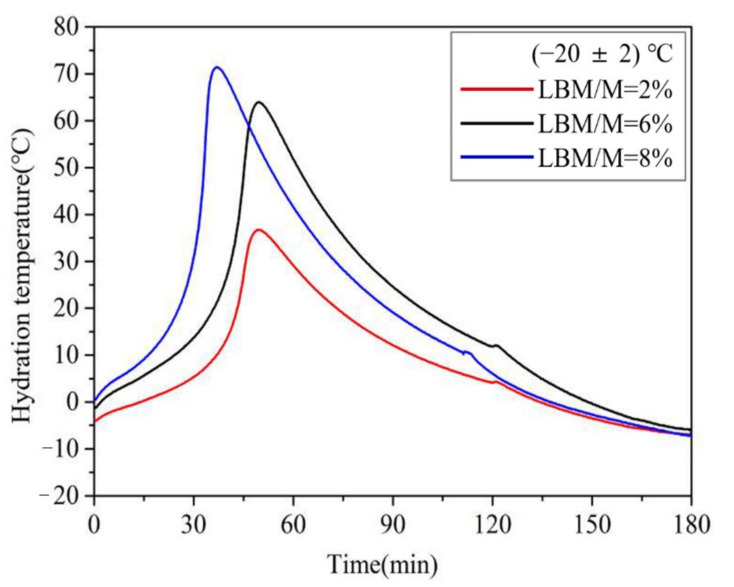
The influences of the LBM/M ratio on the rise in the reaction temperature of MPC at (−20 ± 2) °C.

**Figure 10 materials-13-05587-f010:**
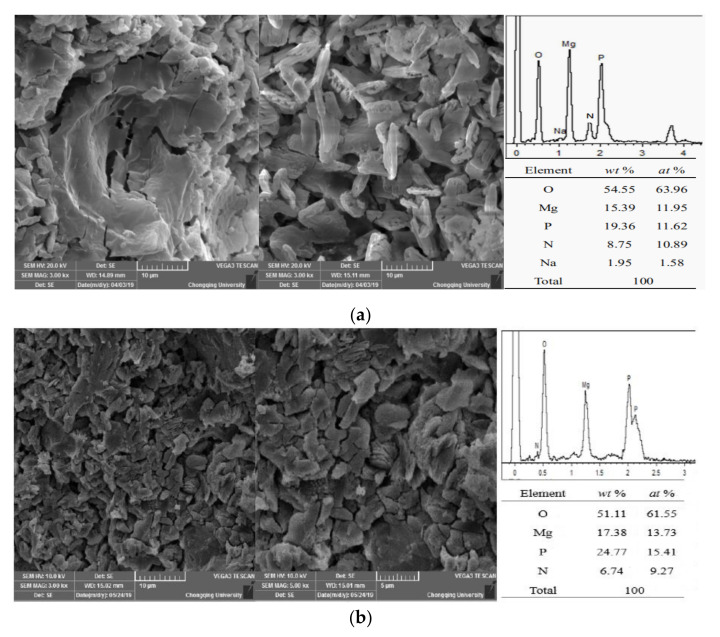
Scanning electron microscopy (SEM) images of MPC samples (1 day) prepared at different temperatures: (**a**) (20 ± 2) °C; (**b**) (−20 ± 2) °C.

**Figure 11 materials-13-05587-f011:**
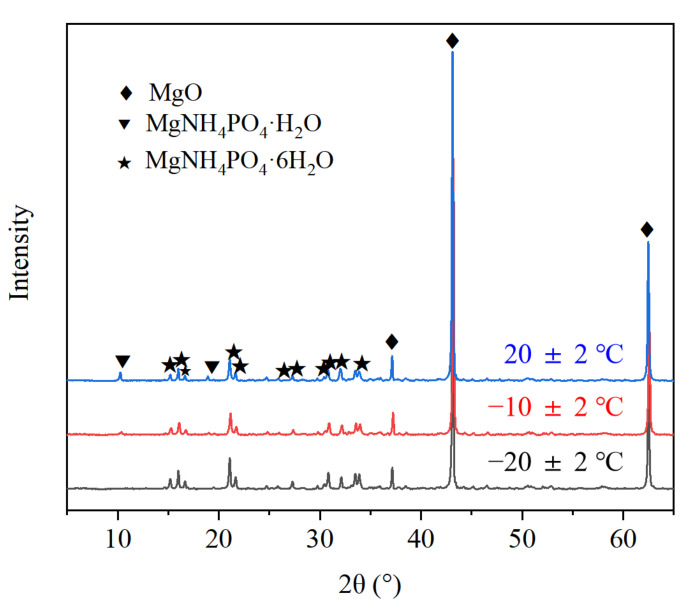
X-ray diffraction (XRD) pattern of MPC (1 day) prepared at different temperatures.

**Figure 12 materials-13-05587-f012:**
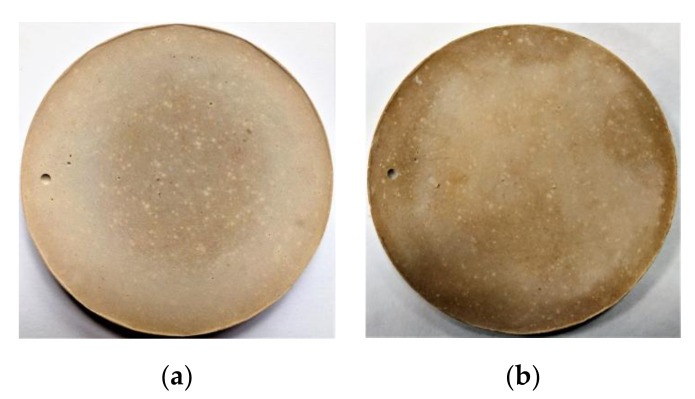
Effect of LBM content on the soundness of MPC: (**a**,**b**) LBM/M = 4%, (**a**) before boiling and (**b**) after boiling; (**c**,**d**) LBM/M = 8%, (**c**) before boiling and (**d**) after boiling.

**Figure 13 materials-13-05587-f013:**
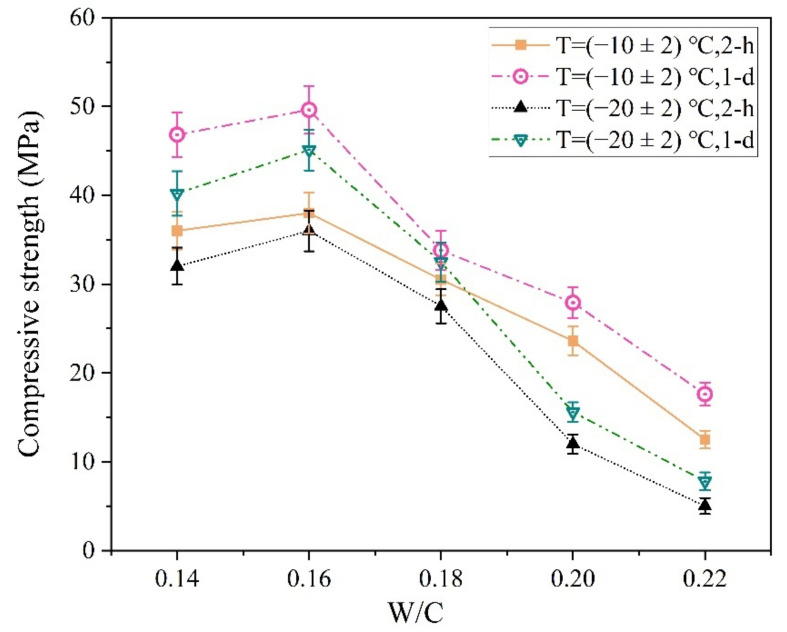
The influence of the W/C ratio on the early strength of the magnesium phosphate cement concrete (MPCC).

**Figure 14 materials-13-05587-f014:**
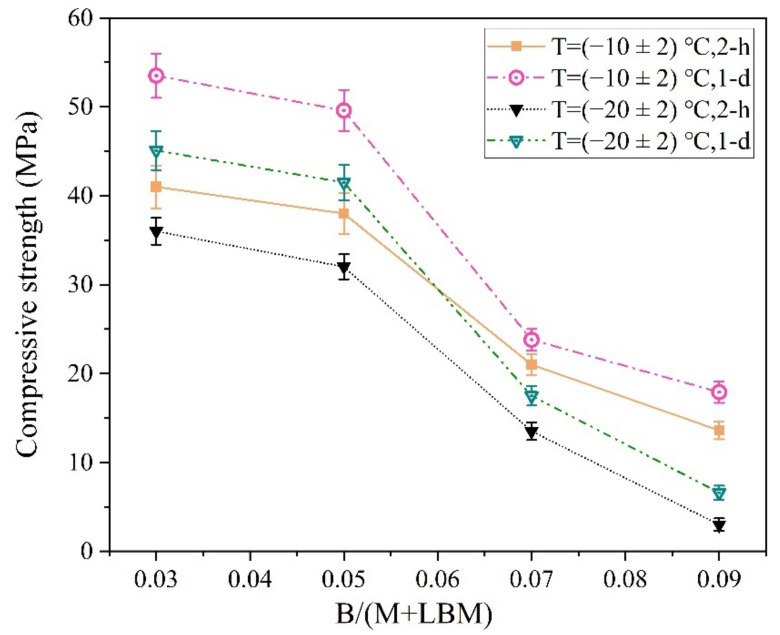
The influence of the B/(M + LBM) ratio on the early strength of the MPCC.

**Figure 15 materials-13-05587-f015:**
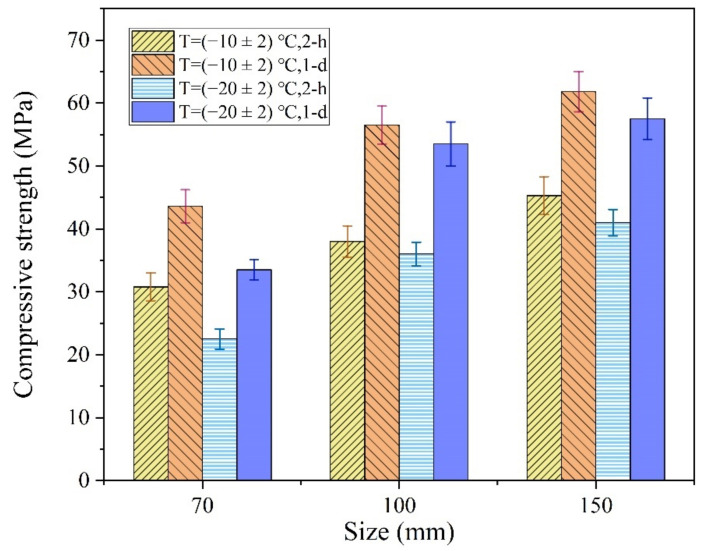
The influences of specimen size on the early strength of the MPCC.

**Figure 16 materials-13-05587-f016:**
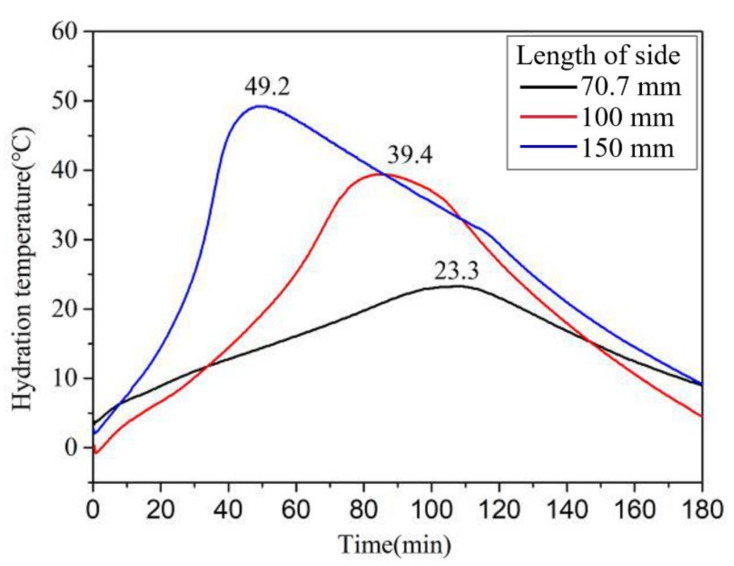
The effect of specimen size on the rises in reaction temperature of the MPCC.

**Figure 17 materials-13-05587-f017:**
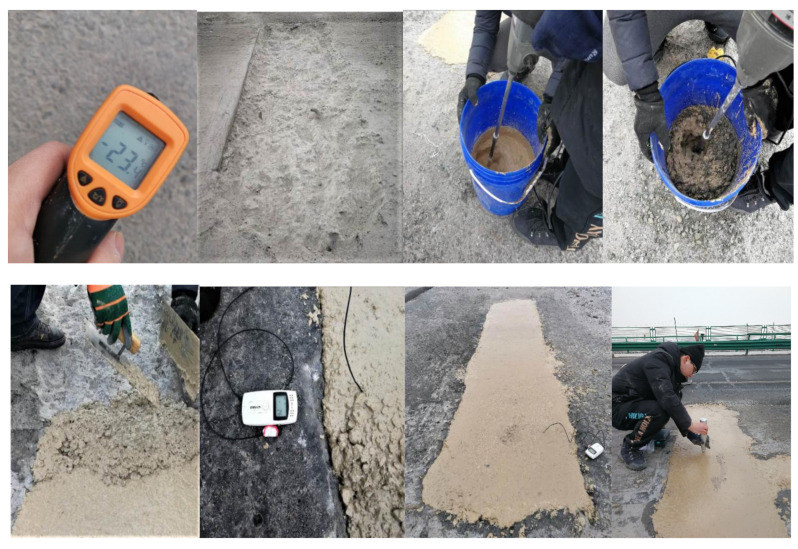
The procedure of rush-repair construction.

**Figure 18 materials-13-05587-f018:**
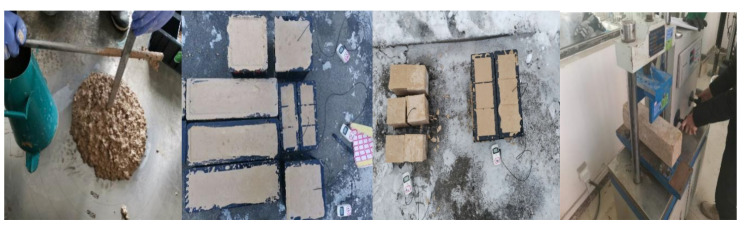
The mixture and specimens of the MPCC.

**Figure 19 materials-13-05587-f019:**
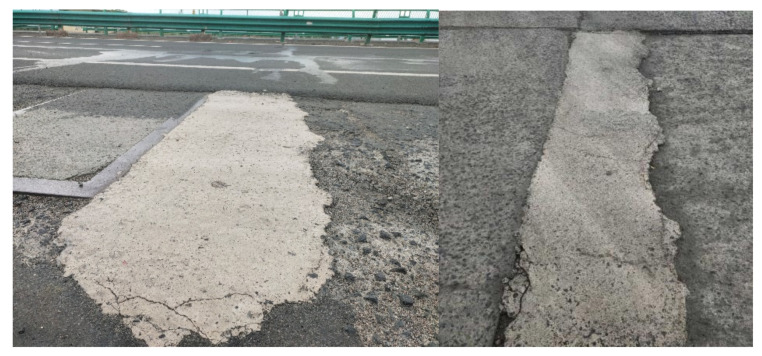
After nine months of service of the repaired pavement.

**Table 1 materials-13-05587-t001:** Chemical composition of dead burnt magnesia (M) and light burnt magnesia (LBM) /%.

Composition	MgO	Si_2_O_3_	CaO	Al_2_O_3_	Fe_2_O_3_	SO_3_	P_2_O_5_	MnO	K_2_O
M	90.78	5.74	2.05	0.65	0.34	0.25	0.05	0.02	0.02
LBM	91.42	5.43	1.98	0.41	0.33	0.26	0.05	0.03	0.02

**Table 2 materials-13-05587-t002:** Early-age performance of magnesium phosphate cement (MPC) prepared and cured at different temperatures.

**Ambient Temperature/°C**	20 ± 2	−10 ± 2	−20 ± 2
**Setting Time/min**	15.0 ^a^	41 ^a^/55 ^b^	68 ^a^/90 ^b^
**2 h Compressive Strength/MPa**	32.0 ^a^	16.0 ^a^/9.5 ^b^	6.0 ^a^/3.5 ^b^

Note: ^a^ the temperature of mixing water was 20 ± 2 °C; ^b^ the temperature of mixing water was 0 °C.

**Table 3 materials-13-05587-t003:** The influences of the mass ratio of MPC/A on the physical and mechanical properties of magnesium phosphate cement concrete (MPCC).

Ambient Temperature	MPC/A Ratio	Setting Time /min	Slump /mm	Compressive Strength/MPa	
				2 h	1 day
T = (20 ± 2) °C B/M = 0.10 LBM/M = 0	1:1	18	210	42.0	57.2
	1:2	21	235	47.0	58.5
	1:3	25	60	31.0	43.3
T = (−10 ± 2) °C B/(M+LBM) = 0.05 LBM/M = 0.04	1:1	20	235	38.0	52.8
	1:2	28	220	33.5	49.6
	1:3	45	90	28.5	40.5
T = (−20 ± 2) °C B/(M+LBM) = 0.03 LBM/M = 0.08	1:1	23	260	36.0	47.5
	1:2	35	210	30.3	45.1
	1:3	50	130	23.5	30.6

Note: MPC = M + LBM + P + B, (M + LBM)/P = 4, W/C = 0.16; A = aggregate (fine and coarse), sand ratio is 0.45.

**Table 4 materials-13-05587-t004:** The ambient temperature measured at the construction site.

07:00	09:00	11:00	13:00	15:00
−23.4 °C	−19.9 °C	−18.5 °C	−17.3 °C	−19.0 °C

**Table 5 materials-13-05587-t005:** The physical and mechanical properties of the MPCC.

Slump	Setting Time	Apparent Density	Maximum Value of Temperature Rises	
210 mm	25 min	2530 kg/m^3^	Center, 36.5 °C	Surface, 1 °C
**2 h Compressive Strength**	**1 d Compressive Strength**	**1 d Flexural Strength**	**2 h Rebound Strength**	**1 d Rebound Strength**
35.0 MPa	54.0 MPa	4.5 MPa	30.0 MPa	48.0 MPa

Note: MPC = M + LBM + P + B; mix proportion: (M + LBM)/P = 4, LBM/M = 0.08, B/(M + LBM) = 0.03, W/C = 0.16, MPC/A = 1:2.

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
