# Peer review of "Preparation and Application of Self-Curing Magnesium Phosphate Cement Concrete with High Early Strength in Severe Cold Environments"

_materials, 2020, doi:10.3390/ma13235587_

Round 1
Reviewer 1 Report
The comments and suggestions are included in the attached file.

Author Response
(1) The authors would like to thank the reviewers for the comments. By taking the reviewers comments on board, References 25 has been added to the Lines 80, and some references already listed in the manuscript have been added in some place of the introduction.
(2) The reviewer suggested that some references should be added in Lines 89-93. The authors would draw the reviewers attention that the purpose of this paragraph is to show the research objectives and contents, therefore, we dose not add reference in this paragraph.
(3) The standard deviation of the test results also has been added on some Figures, for example, Fig.2, Fig.3, Fig.5, Fig.6, Fig.7, Fig.8, Fig.13, Fig.14 and Fig.15.
(4) The standard names in Line 151 and Line 154 have been moved to the reference, at the same time, the other two standard names in Line 144 and Line 149 also have been moved to the reference.
(5) The Fig.14 has been deleted in the updated manuscript.
(6) The standard deviation of has been added on the Fig.15 (the serial number of Fig.16 has been changed to Fig.15 due to the previous Fig.14 has been deleted).
(7) The main factors affecting the slump of magnesium phosphate cement concrete (MPCC) are the W/C ratio, the mass ratio of MPC and the aggregates, especially the W/C ratio. The effects of the mass ratio of MPC and aggregates are shown in Table 3. To demonstrate the effect of W/C ratio on the slump of MPCC mixture, the Fig.14 and Fig.19 show the different slump of MPCC with W/C ratio of 0.14 and 0.18, respectively. The results of this article show that MPCC mixture has good workability (Fig. 19), and the slump of the mixture is very sensitive to the change of W/C ratio (Fig. 14).
Because the main purpose of slump test was to verify the good workability of MPCC mixture, therefore, only the photos of the slump of the MPCC mixture which can meet the pouring requirements were taken during the test, and the picture of MPCC mixture with poor slump were not taken.
(8) The sentence in Line 559 has been revised. The reaction temperature rises of MPC paste increased significantly through the rapid reaction of dilute PA with a small amount of LBM.
Reviewer 2 Report
The research conducted by the authors is very interesting, as it concerns aspects of durability and initial mechanical performance of self-curing magnesium phosphate cement concrete applied in very cold environments. The research therefore offered innovative mixtures to allow greater effectiveness of cementitious mortars used for the repair of structural elements of infrastructures subjected to cold environments. The experimental research is exhaustive for the improvement solutions in the field of restoration cement in the ordinary and extraordinary maintenance programs of infrastructures.
Author Response
Thanks the reviewers for the comments and suggestions. The article has been carefully revised according to the comments.
Reviewer 3 Report
The study is very interesting and only minor changes have to be made.
- At point 3.3.1. (row 451) appear: "The early strength of the MPC decreased gradually with an increase in aggregate content...". Because the aggregates are involved, is better to use MPCC instead of MPC: "The early strength of the MPCC decreased gradually with an increase in aggregate content..."
- At point 3.4 (row 548) appear: "The 2-h compressive strength measured by using a rebound meter was 30.0 MPa, which is about 5.0 MPa lower than that of the MPC specimens." MPC have to be replaced with MPCC because aggregates appear in the recipe.
- At the same point 3.4 (row 548), a comparison is made between the rebound and specimen strength: "The 2-h compressive strength measured by using a rebound meter was 30.0 MPa, which is about 5.0 MPa lower than that of the MPC specimens." Because the rebound test is not very accurate, it is recommended to add a supplementary comment: "Nevertheless, due to errors induced by the rebound method, the real difference can be smaller."
Author Response
Thank the reviewers for your good advice.
- The word MPC has been replaced by the MPCC in lines of 458 and 548.
- The sentence “Nevertheless, due to errors induced by the rebound method, the real difference can be smaller” has been added in lines 545-546.
Reviewer 4 Report
Please see the file attached.

Author Response
Thank the reviewers for your good advice.
- The paper has been checked carefully by a native speaker and some minor errors have been revised
- The samples were treated by spray-gold before SEM test. This introduction has been added in the Section 2.3.
- The stability of MPC needs systematic research under different environment. In this paper, boiling method is mainly used to quickly evaluate the stability of MPC. Thank you very much for the comments and references. Our group will combine micro test to evaluate the stability of MPC repair materials more accurately in the follow-up tests.
- The sizes of the specimens used to test the compressive strength of concrete are mainly 100 mm × 100 mm × 100 mm and 150 mm × 150 mm × 150 mm. The compressive strength values of these two specimens can meet the requirements of engineering application. In addition, there are many factors affecting the size effect of magnesium phosphate cement concrete (MPCC). In this paper, only the influence of environmental temperature on the early strength of MPCC specimens with different sizes was studied. This paper only shows the preliminary test results. Therefore, more systematic test results are needed to evaluate the size effect of MPCC quantitatively and summarize the relevant formulas. The size effect of MPCC is a problem worthy of study, and our team will continue to carry out more systematic research.
Round 2
Reviewer 1 Report
The paper has been improved according to the reviewer's suggestions and it can be accepted in present form.
Author Response
Thank the reviewers for your good advice.